# Targeting the Gut Microbiota: Mechanistic Investigation of Polyphenol Modulation of the Gut–Brain Axis in Alzheimer’s Disease

**DOI:** 10.3390/ijms27020604

**Published:** 2026-01-07

**Authors:** Zhenning Wang, Shanshan Ba, Man Li, Yuanyuan Wei, Yuenan Wang, Jianqin Mao, Yang Xiang, Dongdong Qin, Chuhua Zeng

**Affiliations:** 1School of Basic Medicine, Yunnan University of Chinese Medicine, Kunming 650500, China; wzn7253@163.com (Z.W.); 18388530720@163.com (S.B.); liman09090@163.com (M.L.); wyn18075060922@163.com (Y.W.); xy13197200255@outlook.com (Y.X.); 2First Clinical Medical College, Yunnan University of Chinese Medicine, Kunming 650500, China; wyyuan0000@163.com (Y.W.); mm_0701@foxmail.com (J.M.)

**Keywords:** polyphenols, gut microbiota, gut–brain axis, nanoparticles, Alzheimer’s disease

## Abstract

Alzheimer’s disease (AD) represents an increasingly severe global health challenge. Recently, the role of the gut–brain axis in AD pathogenesis has garnered significant attention. Dysbiosis of the gut microbiota can exacerbate core pathologies such as neuroinflammation, amyloid beta (Aβ) deposition, and tau hyperphosphorylation through neural, endocrine, and immune pathways. Polyphenolic compounds have emerged as a focal point in neuroprotective research owing to their pronounced anti-inflammatory and antioxidant properties. Notably, polyphenols exert effects not only by directly influencing the central nervous system (CNS) but also through indirectly modulating the composition and function of the gut microbiota, thereby impacting bidirectional gut–brain communication. This dual mechanism offers a potential avenue for their application in the prevention and treatment of AD. This review aims to compile recent research on the relationship between polyphenols and the gut microbiota. We assessed the literature from PubMed, Google Scholar, and Web of Science databases, published from the establishment of the database to 24 November 2025. The keywords used include “Polyphenols”, “Gut–brain axis”, “Gut microbiota”, “Alzheimer’s disease”, “Epigallocatechin gallate”, “Quercetin”, “Curcumin”, “Ferulic acid”, “Resveratrol”, “Anthocyanin”, “Myricetin”, “Chlorogenic acid”, etc. This review discusses the various mechanisms by which polyphenols influence AD through modulating the gut microbiota. Polyphenols and gut microbiota exhibit critical bidirectional interactions. On one hand, the bioavailability and activity of polyphenols are highly dependent on metabolic conversion by gut microbiota. On the other hand, polyphenols selectively promote the proliferation of beneficial bacteria such as bifidobacteria and lactobacilli like prebiotics, while inhibiting the growth of pathogenic bacteria. This reshapes the intestinal microecology, enhances barrier function, and regulates beneficial metabolites. Utilizing a nanotechnology-based drug delivery system, the pharmacokinetic stability and brain targeting efficacy of polyphenols can be significantly enhanced, providing innovative opportunities for the targeted prevention and management of AD.

## 1. Introduction

Alzheimer’s disease (AD) is an age-associated neurodegenerative disorder (NDD), with advancing age serving as the primary risk factor for its pathogenesis [1]. Approximately 18.2% of the elderly population aged 65 and above are diagnosed with AD, with prevalence increasing to 33.4% among those aged 85 and older [1]. The global prevalence of AD has been increasing annually. By 2010, the estimated worldwide population of dementia patients had surpassed 55 million [2]. The principal phenotypic presentations of AD encompass gradual neurocognitive deterioration, amnestic deficits, language and visuospatial impairments, as well as neuropsychiatric and personality alterations, eventually culminating in mortality. The principal neuropathological features of AD include the extracellular accumulation of amyloid-beta (Aβ) plaques and intracellular neurofibrillary tangles (NFTs) formed by hyperphosphorylated tau protein, both contributing to neuronal degeneration [1]. Current therapeutic interventions for AD predominantly involve cholinesterase inhibitors, such as donepezil, and N-Methyl-D-Aspartate receptor antagonists, exemplified by memantine. However, these pharmacological interventions only mitigate cognitive deterioration and enhance functionality in activities of daily living; they do not fundamentally inhibit the pathological progression of the disease [3].

The gut–brain axis (GBA) functions as a bidirectional communication network connecting the enteric nervous system (ENS) with the central nervous system (CNS). The main pathways encompass neural routes primarily involving the vagus nerve (VN) and the nucleus tractus solitarius, immunological pathways where microbiota-derived metabolites activate immune cells that subsequently modulate neuroinflammatory processes within the CNS through cytokine and chemokine signaling, and endocrine pathways in which microbiota-secreted compounds directly enter the systemic circulation to influence neuroregulatory functions. The total commensal microbiome within the human colon is estimated to consist of approximately 100 trillion microorganisms [4], encoding over 22 million functional genes [5]. This “second genome” is integral to host metabolic processes, immune system maturation, and neurodevelopmental pathways. The gut microbiome serves as a crucial component of the gut–brain axis, regulating neurodevelopment and modulating synaptic plasticity [6]. Therefore, intestinal microbiota dysbiosis has been identified as a substantial modifiable etiological factor in the pathogenesis of diverse neurodegenerative and neuropsychiatric conditions, such as depression, Parkinson’s disease (PD), and AD [7].

Polyphenols are phenylpropanoid secondary phytochemicals intrinsic to vegetal organisms. Polyphenols are a category of phytochemical secondary metabolites characterized by one or more aromatic benzene rings fused with multiple hydroxyl functional groups [8]. They predominantly consist of two principal phytochemical classes: flavonoids and non-flavonoids [9]. Polyphenols, owing to their distinctive bioactive properties—including antioxidant, anti-inflammatory, neuroprotective, and gut microbiota-modulating effects—are posited to have therapeutic and prophylactic potential in NDD such as AD and PD [10]. Data suggest that polyphenol extracts derived from fruits may mitigate age-related cognitive decline in senior populations [11]. Despite their high dietary prevalence, polyphenols exhibit limited bioavailability in vivo. Predominantly, their absorption is contingent upon biotransformation mediated by gut microbiota, which subsequently amplifies their bioactive potential [12].

The authors’ research expertise has focused on constructing animal models of neuropsychiatric diseases and conducting prevention and treatment with the combination of traditional Chinese Medicine and Western medicine. For example, we have published a review article that focuses on the efficacy of polyphenols on ischemic stroke, Parkinson’s disease. It is emphasized that polyphenols exert neuroprotective effects primarily through inhibiting the production of oxidative stress [13]. Furthermore, the therapeutic benefits of polyphenols on autism-spectrum disorders, anxiety disorders, depression, and sleep disorders were also investigated [14]. This review consolidates existing evidence on the interactions between polyphenolic compounds and the gut microbiota in AD, encompassing the following domains: (I) the modulatory effects of polyphenols on AD pathogenesis; (II) the role of the gut–brain axis in AD progression; (III) microbiota-mediated biotransformation of polyphenols; (IV) polyphenol-induced modulation of commensal and pathogenic microbial populations; (V) microbial metabolites’ impact on intestinal barrier integrity; (VI) nanotechnology-based delivery systems to enhance polyphenol bioavailability and targeted CNS delivery. Collectively, these insights underpin the potential for microbiome-guided, nanomedicine-enhanced polyphenol interventions for precision prophylaxis and therapeutics in AD.

## 2. The Gut Microbiota–Gut–Brain Axis and Its Connection to Alzheimer’s Disease

The human gastrointestinal tract hosts a diverse microbial consortium predominantly consisting of bacteria, accompanied by viruses, fungi, protozoa, and parasitic nematodes, collectively designated as the microbiota [15,16]. The “microbiota–gut–brain axis” has become a central area of investigation in NDD research. Researchers administered intracorpuscular injections of Aβ1-42 oligomers into the gastric mucosa of murine models. Over a 12-month period, they documented retrograde translocation of Aβ peptides from the gastrointestinal epithelium to the CNS. These findings imply that gut-to-brain Aβ oligomer transport may be a critical pathway in the etiopathogenesis of AD and related neuroinflammatory processes [17]. The microbial composition of individuals with AD exhibits significant alterations. A minimum of 11 phyla, 15 classes, 14 orders, 38 families, 137 genera, and 51 species (comprising 9.77%) [18]. Among these, the taxonomic groups *Enterobacteriaceae*, *Lachnospiraceae*, *Ruminococcus*, and the genera *Bifidobacterium* and *Bacteroides* were recognized as exhibiting significant variations in relative abundance [18].

### 2.1. Endocrine Pathway

The impact of bacterial metabolic signaling on CNS functionality is an emerging research focus. Bioactive microbial metabolites, including gamma-aminobutyric acid (GABA), serotonin, histamine, and dopamine, act as neurotransmitters or their precursors, playing a pivotal role in modulating neurochemical pathways involved in emotion, behavior, and cognition [19,20]. *Bacteroides*, *Bifidobacterium* and *Lactobacillus* strains possess glutamate decarboxylase enzymatic genes responsible for the decarboxylation of glutamate to GABA [21,22]. Abnormal levels of gamma-aminobutyric acid are present in the CNS and cerebrospinal fluid of AD patients [23,24]. Animal models demonstrate that reactive astrocytes neighboring Aβ deposits upregulate gamma-aminobutyric acid production, thereby perturbing synaptic plasticity and contributing to deficits in cognitive function and memory [25,26,27]. These findings indicate that microbiota-derived GABA could influence neurodegenerative mechanisms implicated in AD pathology.

Bile acids synthesized by gut microbiota can enhance the permeability of the blood-brain barrier (BBB) [28,29]. Following its direct association with cholesterol in neuronal cell membranes, the amyloid precursor protein (APP) demonstrates increased affinity for the phospholipid monolayer of lipid rafts, thereby facilitating enzymatic processing and the production of Aβ [30]. Short-chain fatty acids (SCFAs), functioning as endogenous ligands, selectively activate the extensively expressed free fatty acid receptors (FFARs) and hydroxycarboxylic acid (HCA) receptors. Through modulation of intracellular signaling pathways involving cyclic adenosine monophosphate (cAMP) and calcium flux, they regulate endocrine hormone secretion, immune responses, and [31]. Interestingly, bacterial amyloid proteins exhibit structural and functional parallels with Aβ, including curli fibers generated by *Escherichia coli*, phenol-soluble modulins produced by *Staphylococcus aureus* [32]. It can promote heterotypic crystallization and aggregation of host amyloid-beta via molecular mimicry [33].

### 2.2. Neurotransmission

The VN, emerging from the medulla oblongata, projects extensively along the gastrointestinal mucosa, facilitating bidirectional communication between the gut and the CNS. This system incorporates diverse chemoreceptors and mechanoreceptors capable of detecting signals generated by the gut microbiota and transmitting them to the CNS [34,35]. Particular strains of *Lactobacillus* influence the gene expression of GABA receptor mRNA within the prefrontal cortex and hippocampal areas; however, this modulatory effect is abolished after vagotomy [36]. VN transection also resulted in reduced neurogenesis and cell survival within the dentate gyrus of the mouse hippocampus [37]. While the VN does not traverse the epithelial barrier [38], intestinal epithelial cells and enteric glial cells located within the mucosal and muscular layers can form direct synaptic junctions with vagal afferent fibers, thereby enabling afferent signal transmission to the CNS [39]. Intestinal glial cells are also capable of modulating the permeability of the intestinal epithelial barrier through neuroglial synapses, relaying alterations in the gut microbiome to the CNS [40]. Therefore, changes in the gut microbiome are believed to potentially disrupt hippocampal-dependent memory processes through VN-mediated pathways, consequently contributing to cognitive impairments associated with dementia [41].

### 2.3. Immunomodulation

The gastrointestinal tract hosts a complex and abundant community of immune effector cells. Interactions between the microbiome and the immune system can trigger the release of pro-inflammatory cytokines and chemokines, thereby affecting systemic neuroimmunological homeostasis [42]. Gut microbiota dysbiosis disrupts intestinal barrier integrity. When the gut-vascular barrier is compromised, microbes, their antigens, and inflammatory mediators infiltrate the systemic circulation [43]. The neuroinflammatory response concomitantly induces BBB disruption, thereby amplifying neuroinflammation and β-amyloid aggregation [44]. Additionally, gut microbiota-derived metabolites modulate the ontogeny, phenotypic differentiation, and activation of microglial cells and astrocytes [45]. Microglia play a critical role in maintaining cerebral homeostasis and mediating neuroprotection against insults, including Aβ and tau aggregation. Dysregulation of microglial activity leads to persistent secretion of chemokines, pro-inflammatory cytokines, and reactive oxygen species (ROS), thereby initiating and amplifying neuroinflammatory responses that exacerbate neuronal degeneration and accelerate the progression of AD pathology [46].

Endocrine, neurophysiological, and inflammatory signaling cascades intricately cross-communicate, collectively orchestrating the bidirectional neuro-gastroenteric axis. SCFAs are capable of penetrating the BBB, modulating neurophysiological processes, and regulating cerebral perfusion, thereby impacting neuroinflammatory responses. Afferent fibers of the VN are capable of directly sensing microbe-associated molecular patterns and metabolic derivatives, relaying nociceptive signals to central pain processing pathways [47]. VN influences neuroinflammatory mechanisms within CNS regions such as the hippocampus by modulating both intestinal and systemic inflammatory responses [48]. In conclusion, the gut microbiome contributes to the development of the GBA via endocrine, neural, and immune pathways, collectively impacting the initiation and progression of AD.

## 3. Polyphenol Alleviates Pathology of Alzheimer’s Disease

Polyphenols are bioactive phytochemicals prevalent in fruits, vegetables, and cereals, demonstrating antioxidant, anti-inflammatory, and metabolic modulation activities. These polyphenols can be classified into flavonoid subclasses and non-flavonoid phytochemicals [9]. Flavonoids comprise subclasses such as flavanols and anthocyanins [49]; non-flavonoids encompass phenolic acids, lignans, and others [50]. Its structural heterogeneity defines its multi-modal therapeutic mechanism in AD management (Figure 1).

### 3.1. Polyphenols Alleviate β-Amyloid Deposition in AD

Polyphenolic compounds potentially attenuate β-amyloid accumulation in AD pathology. Aberrant conformations of APP promote overproduction and aggregation of Aβ peptides, resulting in amyloid plaque formation and tangle development, which ultimately induce neurodegeneration and synaptic dysfunction [3]. Myricetin potentially engages in intermolecular hydrogen bonding with the carbonyl functionalities of β-sheet secondary structures through its hydroxyl moieties, thus impeding Aβ fibrillogenesis [51]. Research demonstrates that myricetin at specific concentrations (10 μM, IC50 = 2.8 μM) can upregulate α-secretase expression and enzymatic activity, inhibit β-secretase (BACE-1) activity, and promote the non-amyloidogenic pathway to diminish amyloid-beta (Aβ) production [52].

Epigallocatechin gallate (EGCG) facilitates non-amyloidogenic pathways through inhibition of the extracellular signal-regulated kinase (ERK)/NF-κB signaling cascade, augmentation of α-secretase enzymatic activity, and downregulation of β- and γ-secretase complexes [53,54]. Moreover, EGCG inhibits the nucleation and fibrillogenesis of Aβ-laden protofibrils through binding interactions with intrinsically disordered peptide conformations [54]. EGCG facilitates the disassembly of mature Aβ protofibrils by degrading their structured aggregates into inert, non-toxic amorphous microaggregates [53].

### 3.2. Polyphenols Inhibit Tau Protein Phosphorylation in AD

Excessive hyperphosphorylation of tau protein significantly contributes to AD pathogenesis. Tau, a microtubule-associated protein primarily localized within neuronal axons, is integral to microtubule polymerization, stabilization, and axonal cytoskeletal integrity, playing a crucial role in axonal development and neuronal maturation [55]. The hyperphosphorylation of tau proteoforms is intricately controlled by various kinases, notably AMP-activated protein kinase (AMPK) and glycogen synthase kinase 3 beta (GSK3β) [56,57].

Quercetin attenuates tau hyperphosphorylation through the modulation of GSK3β enzymatic activity [58]. It concurrently mitigates oxidative stress-triggered apoptosis and caspase-3 activation, primarily through modulation of the MAPK and PI3K/Akt/GSK3β signaling cascades [58,59]. In addition to quercetin, naringenin modulates the PI3K/Akt/GSK3β signaling pathway, thereby attenuating tau hyperphosphorylation [60]. Furthermore, Sonawane et al. [61], elucidated that EGCG exhibits time- and concentration-dependent efficacy in disaggregating preformed tau fibrils and destabilizing their oligomeric assemblies.

### 3.3. Polyphenols Reduce Oxidative Stress Levels in AD

Consequently, oxidative stress is widely recognized as a pivotal contributor to NDD, such as AD [62]. Oxidative stress has been posited as the primary pathogenic trigger initiating tau protein aggregation [63].

Flavonoids mitigate oxidative stress caused by reactive oxygen species via multiple biochemical pathways. One such mechanism involves direct neutralization of free radicals, where the hydroxyl moieties in flavonoids undergo oxidation by reactive species, thereby stabilizing the radicals and diminishing their cytotoxic reactivity [64]. For example, the eight hydroxyl functionalities on EGCG efficiently neutralize reactive oxygen species, thereby exhibiting exceptional antioxidant activity [65] Another antioxidative pathway of flavonoids pertains to the suppression of reactive ROS generation, primarily through the inhibition of critical oxidative enzymes such as xanthine oxidase and the chelation of pro-oxidant metal ions, including Fe^2+^ and Cu^2+^ ions, which catalyze radical formation [66]. Quercetin may promote the phosphorylation and activation of the downstream p53 protein by inhibiting the activity of the ERK1/2 and AKT pathways [67]. Under mild oxidative stress conditions, p53 functions as an antioxidant regulator by transcriptionally activating downstream target genes such as TP53-Induced Glycolysis and Apoptosis Regulator, sestrin1/2, and p21. These proteins mitigate reactive oxygen species (ROS) accumulation by modulating glycolytic flux and preserving mitochondrial integrity. Furthermore, proteins like sestrin1/2 and p21 facilitate the dissociation of Keap1 from Nrf2, thereby enhancing Nrf2-mediated endogenous antioxidant responses and systematically reducing intracellular oxidative stress [68]. Furthermore, quercetin and naringenin upregulate glutathione (GSH) synthesis in hippocampal neurons and stimulate the transcription of key antioxidant enzymes, including superoxide dismutase (SOD), catalase (CAT), and glutathione peroxidase (GPX) [69,70]. In addition to regenerating antioxidant enzymes such as SOD, CAT, and GSH as previously described, myricetin also exhibits inhibitory effects on the Fenton reaction through chelation of transition metal ions like Cu^2+^ and Fe^2+^, thereby attenuating ROS generation and markedly augmenting its antioxidant efficacy [71,72].

Non-flavonoid phytochemicals also exhibit significant antioxidant properties, contributing to the attenuation of oxidative stress and neutralization of free radicals. Specifically, ferulic acid, a phenolic acid, mitigates reactive ROS generation by upregulating SOD enzymatic activity and suppressing malondialdehyde concentrations [73,74]. Butylated ferulic acid—a highly lipophilic ester—has been shown to inhibit Aβ1-42 oligomerization, diminish ROS generation, and concurrently enhance endogenous antioxidant defense by activating the nuclear factor erythroid 2-related factor 2 (Nrf2) signaling pathway [75]. The antioxidative response regulated by Nrf2 represents a vital component of the cellular defense arsenal in mammals. It has been documented that Nrf2 activation can be effectively triggered by natural polyphenolic compounds such as curcumin, EGCG, resveratrol, ferulic acid, puerarin, and quercetin [76].

### 3.4. Polyphenols Alleviate Neuroinflammation in AD

Neuroinflammation, an immune response within the CNS, is identified as a pivotal etiological factor in AD and is perceived as a downstream effect of its neuropathological progression [77]. In age-related chronic neuroinflammatory conditions, glial cells undergo phenotypic polarization towards a pro-inflammatory state, resulting in elevated secretion of cytokines such as IL-1β, IL-6, and TNF-α, which further intensifies neuroinflammatory processes [78]. Clinically, neuroinflammatory responses have been inversely associated with cognitive decline in AD patients [63].

The polyphenolic constituents of pomegranates, including punicalagin and ellagic acid, upregulate IL-10 anti-inflammatory cytokine expression and suppress IL-1β pro-inflammatory cytokine secretion in an Aβ1-42-induced neuroinflammatory model. This modulation appears to be mediated through downregulation of CD86 and upregulation of CD163, promoting the polarization of microglia and macrophages from a pro-inflammatory M1 phenotype to an anti-inflammatory M2 phenotype [79]. Similarly, quinic acid (QA) exhibits neuroprotective effects by mitigating neuroinflammation via suppression of the retinoic acid-induced protein 3/iκB kinase complex/NF-κB signaling cascade, mediated through modulation of gut microbiota-derived metabolites such as indole-3-acetic acid and kynurenic acid [80]. NF-κB, a central transcription factor modulated by diverse inflammation-associated signaling cascades, plays a pivotal role in the regulation of inflammatory responses. Phytochemical polyphenols—including naringin, myricetin, and icariin—demonstrate anti-inflammatory properties through the suppression of NF-κB activation [81,82,83].

### 3.5. Clinical Research on Polyphenol Therapy for Alzheimer’s Disease

We have compiled clinical trial data examining the efficacy of polyphenol-based interventions for AD. Findings demonstrate that polyphenol isolates and extracts derived from specific botanical sources can enhance cognitive function and memory deficits in patients. Furthermore, these compounds facilitate the enhancement of AD-related biomarkers in both serum and cerebrospinal fluid (Table 1). However, certain empirical investigations have demonstrated that polyphenol supplementation does not produce a statistically significant divergence from placebo controls [84,85]. The causes of this paradoxical result may be attributable to various determinants, such as sample population size, patient adherence to therapy, and pharmacokinetic properties of the medication.

**Table 1 ijms-27-00604-t001:** Clinical trials of polyphenols for the treatment of Alzheimer’s disease.

Intervention Drugs	Test Subjects	Test Type	Intervention Time	Cognitive and Memory Functions	AD-Related Biomarkers	References
Spirulina maxima Extract	80 patients with mild cognitive impairment (average age 68.26 ± 4.68 years)	Randomized, placebo-controlled, double-blind study	12 weeks	Visual learning and visual working memory are enhanced. (*p* < 0.05)	BDNF and Aβ showed no significant intergroup differences.	[86]
Resveratrol	119 patients with AD (aged ≥ 49 years)	Randomized, placebo-controlled, double-blind study	52 weeks	There was no significant difference between the treatment group and the placebo group.	Compared with the placebo serum group, the decrease in Aβ40 in cerebrospinal fluid was more pronounced (*p* < 0.05).	[87]
Dietary flavonoids	Healthy elderly individuals aged 60 and above	Randomized, placebo-controlled, double-blind study	3 years	Memory function improved in individuals with low dietary quality and low flavonoid intake. (*p* < 0.01)		[88]
Resveratrol	119 patients with mild to moderate AD	Randomized, placebo-controlled, double-blind, multi-site study	52 weeks	The placebo group exhibited a significant decline in MMSE scores (*p* < 0.01); the resveratrol group showed no significant decline.	Cerebrospinal fluid Aβ42 and Aβ40 levels decreased in both groups. However, the placebo group exhibited a greater downward trend than the resveratrol group (Aβ42 downward trend, *p* = 0.0618). Tau levels showed no significant change.	[89]
Ginkgo biloba leaf extract EGb 761	333 patients with AD (aged ≥ 50 years)	Randomized, placebo-controlled, double-blind study	24 weeks	SKT score: Compared with the placebo group, the treatment group demonstrated improved cognitive function (*p* < 0.001).		[90]

## 4. The Link Between Polyphenols in AD and Gut Microbiota

Polyphenols and their metabolites can extensively enhance gastrointestinal homeostasis by modulating the composition of the gut microbiota, affecting microbial metabolic pathways, and improving intestinal barrier integrity (Figure 2).

### 4.1. Gut Microbiota Influences Polyphenol Metabolism

Polyphenols demonstrate low bioavailability and intricate molecular architectures, which hinder their absorption in the human gastrointestinal system [9]. The intestinal microbiota enzymatically transforms macromolecular polyphenols into smaller metabolites via processes such as deglycosylation, methylation, sulphation, and glucuronidation [91]. These compounds exhibit increased intestinal bioavailability and prolonged plasma residence time, thereby substantially improving the systemic bioavailability of polyphenolic constituents [9].

The biotransformation of polyphenols by gut microbiota is strongly linked to their bioactivity. Firstly, the pharmacodynamic efficacy of certain polyphenols is contingent upon microbial metabolism within the gastrointestinal tract. *Bacteroides ovale* and slow-growing *Eggerella species* are capable of biotransforming catechin and epicatechin, both flavan-3-ol subclasses, into metabolites including 3-hydroxybenzoic acid (3-HBA) and 3-(3′-hydroxyphenyl)propionic acid [92]. Both compounds have been shown to traverse the BBB, inhibit Aβ production, and mitigate neurodegenerative pathology associated with AD [93]. Gut microbiota metabolizes curcumin into tetrahydrocurcumin [94]. Tetrahydrocurcumin demonstrates pharmacological activity comparable to curcumin, with the ability to suppress pro-inflammatory cytokine secretion, including IL-6 and TNF-α, indicating its therapeutic potential in mitigating oxidative stress and neuroinflammatory pathologies [95].

Microbial metabolites of polyphenols may demonstrate enhanced bioactivity relative to parent polyphenols. The polyphenols present in sorghum exhibit notable antioxidative and anti-inflammatory bioactivities. By augmenting cholinergic neurotransmission to facilitate cognitive function and ameliorate deficits. Fermentation of sorghum utilizing a consortium of microbial strains, including *Lactobacillus casei*, *Lactobacillus reuteri*, and *Lactobacillus plantarum*, significantly amplifies its neurocognitive potentiating properties relative to unfermented samples. This enhancement most likely results from fermentation processes increasing the concentration of polyphenolic metabolites with evident anti-inflammatory and antioxidant activities [96].

Interestingly, certain microbial biotransformation products of polyphenols display emergent bioactivities absent in the parent phytochemicals. In APP/PS1 transgenic murine models, the curcumin-derived metabolites demethylcurcumin and bis-demethoxycurcumin have been shown to upregulate the enzymatic activity of the amyloid-beta-degrading protease chymotrypsin, whereas the parent molecule curcumin did not demonstrate this modulatory effect [97]. Resveratrol-3-O-sulfate, a microbial metabolite of resveratrol, exhibits antagonistic activity towards human estrogen receptor subtypes alpha and beta, a property absent in resveratrol itself [98].

Variations in individual microbial community profiles influence the microbial biotransformation pathways of polyphenols. In samples devoid of the *Blautia* sp. strain, the biosynthesis of demethylcurcumin and bis-demethylcurcumin was absent, suggesting that microbial-mediated demethylation of curcumin was not facilitated [99].

Furthermore, specific gut microbiota can biosynthesize select polyphenols independently of dietary intake. For instance, ferulic acid can be efficiently and extensively produced by strains such as *Lactobacillus fermentum* and *Bifidobacterium animalis* [100]. Certain microbial taxa possessing feruloylesterase-encoding genes can catalyze the cleavage of ester linkages to release free ferulic acid from bound forms, thus facilitating its bioavailability and biological activity [100,101]. Ferulic acid enhances activation of the PI3K/AKT signaling cascade, leading to suppression of its downstream effector GSK3β, which subsequently mitigates NFT formation in AD [102].

### 4.2. Polyphenols Modulate the Composition of Gut Microbiota

Dysbiosis presents as an enrichment of pro-inflammatory microbial taxa and depletion of anti-inflammatory commensals, aggravating intestinal inflammation and facilitating β-amyloid aggregation [103]. Polyphenolic phytochemicals can bidirectionally alter microbial community structure, restoring eubiosis through selective antimicrobial activity against pathogenic bacteria or promotion of probiotic proliferation, thus sustaining intestinal homeostasis and mitigating neuropathological features of AD.

#### 4.2.1. Polyphenol-Enhancing Beneficial Microorganisms Associated with AD

Polyphenolic compounds and their metabolites function as prebiotic agents, selectively enhancing the proliferation of advantageous gut microbiota such as *Bifidobacteria* and *Lactobacilli* [104]. The *bifidobacteria*/flavonoid complex markedly suppresses nitric oxide synthesis in lipopolysaccharide (LPS)-activated *RAW264 macrophages* [105]. Bifidobacterium breve modulates immune-inflammatory gene expression, alleviates hippocampal neuroinflammation, and ameliorates β-amyloid-induced cognitive deficits in a mouse model [106]. *Lactobacillus* demonstrates neuroprotective anti-aging effects and mitigates cognitive deficits associated with senescence. In a murine model of D-galactose-induced aging, oral administration of *Lactobacillus plantarum* (strain C29; 1 × 10^10^ CFU) resulted in measurable enhancements in spatial learning and memory performance [107]. Serum concentrations of mRNA and protein expression levels for brain-derived neurotrophic factor (BDNF), cyclic AMP response element-binding protein, interleukin-10 (IL-10), and CD206 were significantly increased. Multiple clinical studies suggest that probiotic formulations containing *Bifidobacterium* and *Lactobacillus* strains can attenuate oxidative stress markers within AD neuropathology and significantly enhance cognitive outcomes [106,108]. Nimgampalle et al. [109], demonstrated that probiotic *Lactobacillus* strains mitigate cognitive impairments in rodent models, decrease β-amyloid plaque accumulation and NTS formation in the hippocampal region, and normalize acetylcholine neurotransmitter levels. Polyphenolic compounds, including curcumin, quercetin, resveratrol, anthocyanins, and myricetin, can modulate the gut microbiota by increasing the relative abundance of *Bifidobacterium* and *Lactobacillus* genera [110,111,112,113,114].

Additional probiotic strains also demonstrate specific correlations with AD pathogenesis and may be modulated by polyphenolic phytochemicals. Butyrate-generating microbiota metabolize dietary fibers into butyrate, with their abundance being strongly correlated to anti-inflammatory, intestinal barrier integrity, metabolic regulation, and neuroprotection. Curcumin demonstrates the potential to mitigate neuroinflammation by promoting the proliferation of these butyrate producers, thereby contributing to the attenuation of AD pathology [110].

Estrogen deficiency and high-fat diets (HFD) are significant risk factors for AD. In relevant mice models, a decreased abundance of *Bacteroides fragilis*, *Lactobacillaceae*, and *Prevotella* species was observed. Following EGCG intervention, cognitive deficits in mice improved, with *Prevotella* abundance significantly restored [115]. These findings suggest that the pathogenesis of AD may be associated with the abundance of *Prevotella* species, and that EGCG exerts its effects by remodeling the gut microbiota, suppressing inflammation, and enhancing gut–brain axis function.

In another study assessing the impact of kaempferol on the gut microbiota, kaempferol intervention was observed to increase the *Firmicutes-Bacteroidetes* ratio in AD patients, enhance the abundance of beneficial taxa such as *Prevotellaceae* and *Ruminococcaceae*, while concurrently inhibiting the growth of pathogenic bacteria [116]. These modifications are intricately linked to colitis pathophysiology and intestinal epithelial barrier integrity.

*Akkermansia* exhibits therapeutic potential as a probiotic in mitigating diverse pathological conditions, attenuating gut mucosal inflammation, and preserving intestinal barrier integrity [117]. A research investigation examining the impact of QA on HFD-induced neuroinflammatory responses and its prospective role in AD mitigation demonstrated that QA markedly elevated the relative abundance of *Akkermansia*, implying that this microbial genus may be integral to QA’s modulatory effects on neuroinflammation pathways [80]. Wu et al. [118], identified that fecal microbiota transplantation from EGCG-administered murine donors facilitated the expansion of advantageous microbial taxa, notably *Akkermansia*. These microbiota-driven metabolic and immunological modifications not only sustain gastrointestinal homeostasis but also modulate AD pathology by attenuating neuroinflammation and Aβ accumulation via the GBA.

#### 4.2.2. Polyphenols Inhibit the Proliferation of AD-Associated Pathogenic Bacteria

AD patients demonstrate significant dysbiosis characterized by decreased levels of commensal gut microbiota and proliferation of pathogenic microbial species. Evidence suggests that increased abundance of *Escherichia coli* and *Shigella* spp. is positively associated with systemic inflammatory markers in individuals exhibiting cognitive decline and cerebral amyloid accumulation [119]. Curcumin mitigates cognitive impairments and neuropathological changes in a murine model of AD. It concurrently modulates the gut microbiome by suppressing the proliferation of *Escherichia coli* and *Shigella species* [97]. Furthermore, curcumin significantly reduces the abundance of pathogenic bacteria such as *Prevotella*, *Corynebacterium*, *Enterobacter*, and *Ruminococcus* [120,121,122]. Akbari et al. [96], demonstrated that polyphenolic compounds in red sorghum significantly reduced the abundance of *Escherichia coli* in the intestines of AD rats, thereby exerting therapeutic effects on AD. *Helicobacter pylori* has been established as a critical pathogenic contributor to AD. Empirical evidence indicates that, under pathological states, it can transgress the blood-brain barrier [123]. Notable elevations in *Helicobacter pylori* colonization levels were observed in AD murine models [124], which is strongly correlated with heightened activation of oxidative stress pathways [125].

The antimicrobial mechanisms of polyphenols encompass four interconnected pathways: enzymatic inhibition, cellular wall destabilization, membrane lipid disruption, and induction of metabolic nutrient deprivation [126]:

Firstly, polyphenols specifically target pivotal bacterial enzymes, including dihydrofolate reductase and DNA gyrase, thereby disrupting critical metabolic pathways such as tetrahydrofolate biosynthesis, fatty acid anabolism, DNA supercoiling regulation via topoisomerase, and ATP generation [127]. As these enzymes are exclusively present in prokaryotes, and the active sites of functionally similar enzymes in bacteria and human cells exhibit significant structural differences [128,129,130], polyphenols can target bacteria with precision while exerting minimal effects on the human host. Secondly, polyphenolic hydroxyl functionalities have been shown to impede bacterial peptidoglycan biosynthesis and compromise cell membrane integrity [131]. For instance, chestnut bur polyphenol extract (CBPE) can compromise the integrity of the cell envelope in *Shigella dysenteriae*, resulting in a marked elevation of alkaline phosphatase (AKP) activity [132]. Thirdly, polyphenolic compounds can compromise the integrity and selective permeability of lipid bilayer membranes. For instance, ellagitannins have been shown to disrupt membrane permeability in *Escherichia coli* [133]. Bergamot polyphenols suppress bacterial proliferation by disrupting the biosynthetic pathways involved in cell membrane assembly [134]. Finally, polyphenols can provoke microbial nutrient starvation; for example, EGCG obstructs glucose uptake, while gallotannins inhibit microbial proliferation through chelation of vital minerals like iron [135].

The antimicrobial efficacy of polyphenols is critically modulated by their molecular architecture, notably the presence of functional groups such as -OH, -OCH_3_, and glycosylation moieties. The hexahydroxydiphenol and nonahydroxytriphenol contained within ellagic acid tannins, due to their abundant -OH groups, exhibit significant antibacterial activity against *Staphylococcus aureus* and *β-hemolytic streptococci* by chelating metal ions and disrupting cell membranes [133,136]. Secondly, the substitution position and density of -OCH3 groups selectively modulate the antimicrobial activity spectrum: 5,7-dihydroxy-4′,6,8-trimethoxyflavone demonstrates efficacy against *Staphylococcus aureus*, *Escherichia coli*, and *Streptomyces* spp., whereas 5,6-dihydroxy-4′,7,8-trimethoxyflavone exhibits negligible activity against these pathogens but shows potent antifungal activity against *Bacillus subtilis* and *Candida albicans* [137].

Glycosylation modulates the antimicrobial efficacy of polyphenolic compounds bidirectionally: conjugation of kaempferol and quercetin with rhamnose or arabinose at the C-3 hydroxyl group enhances their activity against bacterial pathogens, including *Staphylococcus aureus* [133]. However, glycosylation of the phenolic hydroxyl moiety in resveratrol markedly diminishes its antimicrobial efficacy [138]. QA and shikimic acid derivatives display heightened hydrophilicity resulting from glycosylation and carboxylation, which perturb the hydrophilic-lipophilic balance and compromise their membrane permeation, consequently attenuating their antimicrobial efficacy. However, glucose-conjugated derivatives did not demonstrate notable alterations in antimicrobial efficacy, implying that the influence of glycosylation on polyphenol’s antibacterial properties may be ligand-specific, potentially involving sugars such as rhamnose or arabinose [133].

The antimicrobial potency of polyphenolic compounds is modulated by concentration, exposure time, and environmental parameters. The bacteriostatic activity of betel nut seed-derived polyphenols against microbial strains such as *Escherichia coli* demonstrates a dose-dependent response within the concentration range of 10–50 mg/mL, with peak efficacy observed at 50 mg/mL [139]. Furthermore, the antimicrobial efficacy of baicalin against *Salmonella* spp. and *Staphylococcus aureus* is attenuated under hyperglycemic conditions, suggesting that environmental factors influence the bioactivity of polyphenolic phytochemicals [138].

#### 4.2.3. Research into Polyphenols and the Gut Microbiota in Alzheimer’s Disease

In conclusion, polyphenols confer neuroprotective effects in AD by modulating the gut microbiota, promoting the proliferation of symbiotic bacterial species, and suppressing the growth of pathogenic microbial populations. We have synthesized research assessing the impact of specific polyphenols on gut microbiota composition within AD experimental models (Table 2). The findings indicate that various polyphenolic compounds impose similar modulatory influences on specific microbial consortia. *Lachnospiraceae*, *Akkermansia* are pivotal probiotic taxa linked to AD pathology, with polyphenolic compounds demonstrated to augment their relative abundance within the gut microbiota across diverse AD models. For instance, anthocyanins, resveratrol, and quercetin are capable of augmenting the relative abundance of *Lachnospiraceae*, whereas quercetin, protocatechuic acid, and QA can similarly elevate the prevalence of *Akkermansia* within the gut microbiota. This indicates that certain microbial communities play a key role in the pathogenesis and treatment of AD. Interestingly, the same polyphenol exhibits differing effects on the microbiota when applied to distinct AD models. For example, quercetin induced differential microbiome alterations in HFD-fed mice subjected to intraperitoneal D-galactose administration, relative to age naturally aged controls. Related studies on polyphenols such as resveratrol further corroborate this observation. These findings suggest that alterations in the gut microbiome constitute a multifaceted challenge, with their final impact potentially modulated by various determinants such as pharmacological interventions, pathophysiological classifications, disease models (which may also serve as etiological variables), and environmental conditions. This complexity must be carefully considered in future microbiome-focused studies.

**Table 2 ijms-27-00604-t002:** Effects of polyphenols on gut microbiota abundance in an AD model.

Polyphenol Name	AD Model	Effects on the Abundance of Relevant Microbial Communities	Neuroprotective Effects on AD	Signaling Pathway	References
Anthocyanin	12-month-old male C57BL/6J aged mice	Abundance increase: *Lachnospiraceae*, *Clostridia*	Regulate inflammatory mediators such as IL-1β and IL-6 to alleviate neuroinflammation.	MAPK signaling pathway	[140]
Curcumin	APP/PS1 Transgenic Mouse	Decreased abundance: *Prevotellaceae* Increased abundance: *Bacteroides*	-	-	[97]
Curcumin	3xTg-AD mice	Abundance decrease: *Verrucomicrobia*. Abundance increase: Family level: *Oscillospiraceae* and *Rikenellaceae* Genus level: *Oscillibacter*, *Alistipes*, *Pseudoflavonifractor*, *Duncaniella*, and *Flintibacter*.	-	-	[141]
EGCG	HFD + OVX mice	Decreased abundance: *Porphyromonadaceae* Increased abundance: Phylum level: *Micrococci*; Family level: *Bacteroideae*, *Rhabdomonadaceae*; Genus level: *Prevotella*, *Prevotella*-like	Iron Intake and Redox Homeostasis	-	[115]
Resveratrol	HFD + AlCl_3_ (50 mg/kg) + D-galactose (120 mg/kg) ICR male mice	Decreased abundance: *Rikenella*, *Anaerotruncus*, *Colidextribacter*, *Helicobacter* Increased abundance: *Lactobacillus*, *Bifidobacterium*, *Allobaculum*, *Alloprevotella*, *Candidatus Saccharimonas*, *Alistipes*, *Parasutterella*	Oxidative Stress and Neuroinflammation	-	[114]
Resveratrol	Oral administration of AlCl_3_ (35 mg/kg) + intraperitoneal injection of D-gal in male ICR mice	Reduced abundance: *Alistipes*, *Odoribacter*, *Helicobacter* Increased abundance: *Desulfovibrio*, *Candidatus Saccharimonas*, *Lachnoclostridium*, *Enterorhabdus*, *Faecalibaculum*	Neuroinflammation	NF-κB/MAPK/Akt signaling pathway	[124]
Resveratrol	HFD + AlCl_3_ (50 mg/kg) + D-galactose (120 mg/kg) ICR male mice	Reduced abundance: *Anaerotruncus Rikenella* Increased abundance: *Desulfovibrio Candidatus*_*Saccharimonas Roseburia Lachnospiraceae*_UCG-006 *Alloprevotella Ruminococcus*	Neuroinflammation	-	[142]
Quercetin	HFD+ Intraperitoneal injection of D-galactose (300 mg/kg) ICR male mice	Reduced abundance: *Firmicutes* Increased abundance: *Akkermansia*, *Lactobacillus*, *Bacteroides*, *Alistipes*, *Lachnospiraceae*_NK4A136_group	Regulate IL-1β and TNF-α to suppress inflammation	-	[143]
Quercetin	15-month-old naturally aged ICR mice + high dAGEs diet	Abundance Decline: Phylum Level: *Proteobacteria*, Tenericutes; Genus Level: *Prevotella*	Regulate miR-219, miR-15a, and miR-132-related factors to suppress neuroinflammation.	ERK1/2 signaling pathway	[144]
Baicalin	APP/PS1 Transgenic Mice	Decreased abundance: p_*Firmicutes* and p_*Proteobacteria* Increased abundance: g_*Lactobacillus*, g_*Bifidobacterium*, g_*Clostridium*	-	-	[145]
Protocatechuic acid	FAD transgenic (Tg) mice	Decreased abundance: *Clostridium*, *Proteobacteria* Increased abundance: *Akkermansia*, *Lactobacillus garnerii*	Reduced expression of pro-inflammatory factors (IL-6, IL-1β, TNF-α) and increased expression of the anti-inflammatory factor IL-10.	NF-κB signaling pathway	[146]
Quinic acid	HFD + C57BL/6 mice.	Decreased abundance: *Verrucomicrobiota* and *Bacteroidetes* at the phylum level; *Colidextribacter*, *Roseburia*, *Blautia*, and *Lanchnospiraceae* at the genus level Increased abundance: *Firmicutes* and *Desulfobacteria*; *Akkermansia* at the genus level	Downregulated expression of AD-related genes (APP, PS1, APH1, APOE, IDE, BACE1). Suppressed neuroinflammation: Reduced expression of pro-inflammatory factors (IL-6, IL-1β, TNF-α) and increased expression of the anti-inflammatory factor IL-10.	DR3/IKK/NF-κB signaling pathway	[80]

### 4.3. Polyphenols Can Regulate Microbial Metabolites

Gut microbial metabolites, such as SCFAs, tryptophan derivatives, and secondary bile acids, play a pivotal role in host immune modulation and gut–brain axis signaling. They can influence blood-brain barrier integrity, suppress neuroinflammation, and modulate cognition-associated histone acetylation [147].

SCFAs including acetate, propionate, and butyrate, are produced by various gut microorganisms (such as *Rosaceae*, *Faecalibacterium*, *Bifidobacterium*, *Lactobacillus*, and *Enterobacter*) through the conversion of complex carbohydrates. By engaging multiple G protein-coupled receptors, SCFAs modulate host immune modulation, inflammatory signaling pathways, and neurotransmitter biosynthesis [148]. Free fatty acid receptor 3 has been identified as being expressed within the peripheral nervous system and the BBB [149]. Polyphenolic compounds may influence short-chain fatty acid levels by promoting the proliferation of specific microbial populations. For instance, EGCG can upregulate *Akkermansia* species and the synthesis of propionic and butyric acids [150], resveratrol increases the abundance of short-chain fatty acid-producing bacteria [151]. Kaempferol also enhances the expression of genes associated with butyric acid synthesis, thereby augmenting the microbial community’s capacity for butyric acid production [152].

As a critical amino acid, tryptophan undergoes enzymatic and microbiota-mediated catabolism within the gastrointestinal tract to generate indole and its metabolic derivatives. These compounds engage in pathways associated with inflammatory responses, redox homeostasis, and mucosal barrier function by activating the aryl hydrocarbon receptor (AhR) and pregnane X receptor within intestinal epithelial cells [153,154]. Certain indole-based compounds possess the ability to traverse the BBB, thereby modulating neuronal excitability and glial cell reactivity [153,155]. *Bacteroides* spp. have been shown to exhibit tryptophan metabolic competence, facilitating the conversion of tryptophan to indole-3-acetic acid in the gastrointestinal microbiota. This metabolic activity modulates host innate immunity and mucosal barrier integrity via activation of the AhR signaling pathway [156].

Bile acid analogs engage G protein-coupled bile acid receptor 1 within neuronal and glial populations, downregulating mitochondrial biogenesis and reducing reactive oxygen species production, thereby attenuating NDD progression [157,158]. Naringin administration markedly elevated the relative abundance of the genus *Bacteroides*, an obligate anaerobic microbial taxon known to produce 7α-dehydrogenase enzyme. This enzyme catalyzes the dehydroxylation at the C-7 position of primary bile acids, resulting in the formation of secondary bile acids; this process contributes to the expansion of the secondary bile acid pool and facilitates the activation of associated receptor-mediated signaling pathways [159].

### 4.4. Polyphenols and Microbiota Modulate the Intestinal Barrier

The intestinal barrier consists of a mucus layer, epithelial cells, and immune cells, with its integrity reliant on microbial homeostasis [160]. Microbiome dysbiosis can impair barrier integrity, permitting microbial metabolites and inflammatory mediators to translocate into the circulatory system and trigger neuroinflammation [161].

The mucus layer, composed predominantly of mucin 2 (MUC2), isolates the intestinal lumen from the epithelium. Its synthesis is regulated by the gut microbiota and is closely associated with the composition and activity of the intestinal microbiome [162,163]. The mucous barrier can additionally establish several “bacteria-sparse” zones, thereby further decreasing the probability of pathogenic microorganisms reaching the colonic epithelium [164]. MUC2 demonstrates an initial compensatory upregulation, succeeded by a progressive decline over the course of AD progression [165]. When AD is combined with Irritable Bowel Syndrome symptoms, an increase in the abundance of mucolytic bacteria is observed [166]. Polyphenolic phytochemicals have been demonstrated to upregulate the transcriptional activity of the MUC2 gene [167]. Furthermore, *Akkermansia* muciniphila—a pivotal mucin-degrading commensal—plays a critical role in maintaining intestinal mucosal barrier function through mucin catabolism, leading to mucus layer homeostasis. Dietary supplementation with proanthocyanidins markedly elevates their relative gut microbiota abundance, potentially contributing to enhanced barrier integrity [168]. This observation indicates that the mechanism underlying proanthocyanidins’ enhancement of intestinal barrier integrity likely involves their regulatory influence on mucin biosynthesis and their facilitation of mucus layer restoration.

Tight junction proteins in intestinal epithelial cells, such as zonula occludens-1 (ZO-1), desmoglein, and occludin, structurally integrity of the intestinal epithelial barrier via cytoskeletal anchoring and intercellular adhesion, thereby effectively restricting the paracellular translocation of pro-inflammatory mediators, pathogens, and deleterious substances through both physical occlusion and functional regulation mechanisms [169].

Certain polyphenolic compounds can modulate intestinal barrier integrity through direct regulation of tight junction protein expression. For instance, anthocyanins and baicalin reinforce intestinal barrier function by upregulating the expression of tight junction proteins (Occludin, Claudin, ZO-1) [170,171]. Interestingly, microbiota-derived metabolites generated during anthocyanin biotransformation may also confer advantageous effects on intestinal epithelial integrity. Resveratrol-3-O-sulfate enhances the transcriptional and translational levels of tight junction constituents such as occludin, ZO-1, claudin-1, and claudin-4 in Caco-2 epithelial monolayers, thereby preserving intestinal epithelial barrier function [172]. This suggests that polyphenolic metabolites may contribute to the preservation of intestinal barrier integrity through modulation of tight junction proteins.

## 5. Enhancing Polyphenol Bioavailability Through Nanodelivery Technology

Polyphenolic compounds provide neuroprotection through the modulation of gut microbiota; however, their clinical application is limited by poor bioavailability. This primarily arises from their intricate chemical structures, poor aqueous solubility, hydrophobic nature, and vulnerability to hydrolytic degradation within the gastrointestinal environment, leading to suboptimal absorption and limited systemic availability [173].

To address the aforementioned issues, nanocarrier systems have emerged as a significant strategy. Such systems typically refer to solid colloidal drug delivery nanocarriers, with at least one dimension within the nanoscale range (1–100 nanometers). They are primarily categorized into organic nanoparticles—including natural polysaccharides, lipids, and proteins—and inorganic nanoparticles, such as metal and metal oxide nanoparticles, carbon-based nanomaterials, and calcium-based nanostructures [174]. Their nanometric scale and extensive surface area enhance optimal drug encapsulation and precise targeting, while concurrently accommodating both hydrophilic and hydrophobic agents [175]. Nanoencapsulation safeguards polyphenols against enzymatic hydrolysis in the gastric environment, facilitating controlled-release and targeted delivery within the gastrointestinal tract. This markedly improves their bioaccessibility and maximizes their modulation of the gut microbiome [124,142,143].

Polyglutamic acid is a water-soluble, biodegradable multivalent polypeptide with high loading capacity. Bisdemethoxycurcumin conjugated with spherical crosslinked self-assembled star-shaped polyglutamic acid demonstrates efficient BBB penetrance and CNS cellular uptake, mitigating Aβ-induced neurotoxicity in transgenic AD models and promoting neurotrophic activity in hippocampal organoid systems [176]. Furthermore, β-lactoglobulin-EGCG nanoencapsulates demonstrate a notable resilience to gastric fluid, rendering them effective as natural bioactive compound delivery systems for sustained release of EGCG [177]. In addition to protein carriers, polysaccharide matrices can critically improve polyphenol bioavailability: the binding of polyphenols to plant cell wall components directly affects their release kinetics and intestinal absorption. Positively charged chitosan, owing to its low cytotoxicity, biodegradability, and excellent biocompatibility, is extensively employed in the synthesis of polyphenol-chitosan conjugates. These conjugates significantly improve mucoadhesive properties within the gastrointestinal epithelium, facilitating targeted, controlled release of polyphenolic compounds [173]. An experimental study in AD murine models revealed [142] that resveratrol encapsulated within selenium and chitosan nanoparticle delivery systems modulates the composition of gut microbiota, including *Enterococcus* spp., *Escherichia coli*, *Rickettsia* spp., and *Candidatus Saccharimonas*—microbial taxa implicated in oxidative stress, inflammatory responses, and lipid accumulation. This methodology attenuated LPS secretion and neuroinflammatory responses, concomitantly modulating gut microbiota by decreasing *Firmicutes* and elevating *Bacteroidetes* populations. It also mitigated tau hyperphosphorylation and Aβ deposition within neural tissue through the modulation of the c-jun N-terminal kinase/AKT/GSK3β signaling cascade. Additionally, the evidence indicates that intranasal delivery of lipid nanostructures markedly surpassed oral routes in therapeutic outcomes. These findings suggest that intranasally administered lipid nanocarriers exhibit considerable promise regarding treatment efficacy for AD, safety profiles, and patient adherence [178].

Within the field of inorganic nanomaterials, selenium-core nanoparticles have been predominantly examined in AD pathogenesis studies. For example, Li et al. [124], developed selenium nanoparticle-based nanodelivery systems (TGN-Res@SeNPs) capable of traversing the BBB. Their findings demonstrated that these nanocarriers inhibit Aβ aggregation, attenuate neuroinflammation via modulation of NF-κB, MAPK, and Akt signaling pathways, reduce ROS levels, and remodel oxidative stress-associated microbiota—including *Alistipes* and *Helicobacter* species—leading to a significant decrease in hippocampal Aβ accumulation and enhancement of cognitive performance. Similarly, chlorogenic acid encapsulated within selenium nanocarriers (TGN-CGA@SeNCs) demonstrates comparable bioactivity, effectively enhancing the relative abundance of probiotic genera such as *Turicibacter* and *Ruminococcus* [179]. Both nanoplatforms demonstrate dose reduction of therapeutic agents, improved bioavailability, and multifaceted mitigation of AD pathology via a tripartite mechanism involving BBB penetration, intracerebral targeting, and modulation of gut microbiota. This constitutes an innovative nanotherapeutic strategy for precision brain intervention in AD.

Nano-delivery platforms facilitate enhanced bioavailability of polyphenolic phytochemicals in AD therapeutics, markedly improving their physicochemical stability, solubility, and cellular uptake, thus augmenting their pharmacodynamic efficacy in metabolic pathologies such as AD. Nonetheless, further optimization of formulation methodologies is required, along with comprehensive safety assessments and longitudinal clinical trials to substantiate the enduring therapeutic efficacy in AD pathology. Collectively, this strategy establishes a robust framework for fully harnessing the neuroprotective potential of polyphenolic compounds in the management of AD.

## 6. Discussion

### 6.1. Mechanisms of Polyphenol–Microbiota Interactions in AD

This review systematically synthesizes current evidence on the interactions between polyphenolic compounds and gut microbiota in AD, highlighting the pivotal role of the intricate “polyphenol–microbiota–gut–brain axis” network in the pathophysiological mechanisms of AD. Polyphenols do not function as standalone bioactive compounds; their pharmacological efficacy is predominantly contingent upon reciprocal interactions with the host gut microbiota. Structurally intricate polyphenols necessitate microbial bi-otransformation processes—such as deglycosylation—to produce metabolites with improved bioavailability, augmented bioactivity, or entirely novel chemical entities (e.g., tetrahydrocurcumin) [94]. Conversely, polyphenols modulate the composition of the gut microbiome, selectively enhancing the growth of commensal microbial taxa such as *Bifidobacteria*, *Lactobacillus* spp., and *Akkermansia muciniphila* [105,106,180] andattenuating the proliferation of pathogenic microorganisms, including *Escherichia coli*, *Shigella* species and *Helicobacter pylori*, via multi-modal mechanisms such as enzyme inhibition and disruption of microbial membrane integrity [96,97,132]. This re-configuration of the microbiome composition subsequently impacts concentrations of critical metabolites, including SCFAs, bile acids, and neuroactive compounds such as GABA. Via immune, neuroanatomical, and endocrine signaling pathways, it ultimately influences Aβ aggregation, tau protein hyperphosphorylation, neuroinflammatory responses, and oxidative stress within the CNS [31,99,181].

### 6.2. Research Gaps and Limitations

However, as mentioned earlier, fungi and viruses are also components of the gut microbiota. The interactions between bacteria and fungi play a role in maintaining the balance of microbial communities [182]. Unfortunately, we have found that research on gut fungi in AD remains relatively scarce compared to that on bacteria, primarily focusing on *Candida*, *Saccharomyces*, and *Aspergillus* species [183]. Moreover, in studies investigating the gut microbiota mechanisms of polyphenols in AD, whether in research articles or review papers, the primary focus has predominantly been on gut bacteria. Gut fungi and even viruses are often only briefly mentioned without in-depth discussion, or not addressed at all.

Additionally, to address the issues of low polyphenol activity and bioavailability, nanoscale systems have been introduced. However, in the field of AD, research on nanotechnology-related polyphenols remains incomplete. Current studies primarily focus on selenium-based and protein-based nanocarriers [124,176,177,178,184].

### 6.3. Future Directions

Therefore, to more effectively investigate the mechanisms through which polyphenols facilitate the restoration of gut microbiota homeostasis in AD, it is essential to comprehensively include microorganisms such as bacteria, fungi, and viruses as subjects of research. Investigate how polyphenols influence gut bacteria, fungi, and viruses, and examine the roles these factors play in maintaining gut homeostasis.

Moreover, the regulation of microbial abundance by polyphenols exhibits significant variations across different disease models. Future research and therapeutic strategies must account for the unique gut microbiome characteristics of individual hosts. Distinguishing differences between animal and human gut microbiota is essential for achieving better clinical therapeutic outcomes. Developing polyphenol-based nanomedicines from diverse materials for AD, whilst thoroughly investigating their therapeutic advantages and limitations.

Finally, the existing evidence primarily originates from animal models, necessitating the design of comprehensive randomized controlled trials to substantiate the therapeutic effectiveness and safety profile of these interventions in AD patients.

## 7. Conclusions

In conclusion, this research illustrates that integrating polyphenol-based therapy with gut microbiota modulation, augmented by nanotechnology to optimize bioavailability, constitutes a multi-faceted, systemic approach for the prophylaxis and management of AD. Advancing our comprehension of the intricate interplay among polyphenols, microbiota, and host physiology may reveal novel therapeutic pathways for AD intervention.

## Figures and Tables

**Figure 1 ijms-27-00604-f001:**
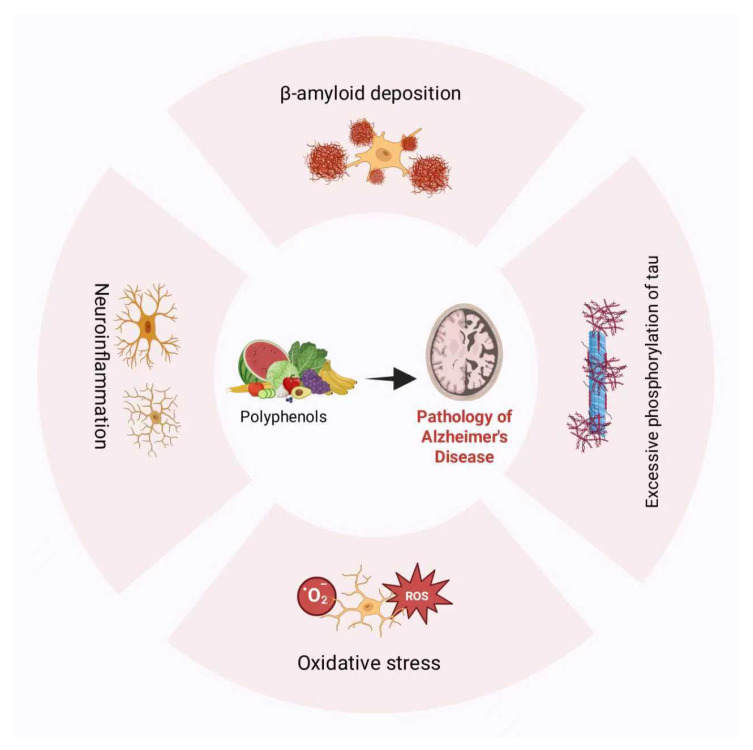
Polyphenol alleviates the pathology of Alzheimer’s disease.

**Figure 2 ijms-27-00604-f002:**
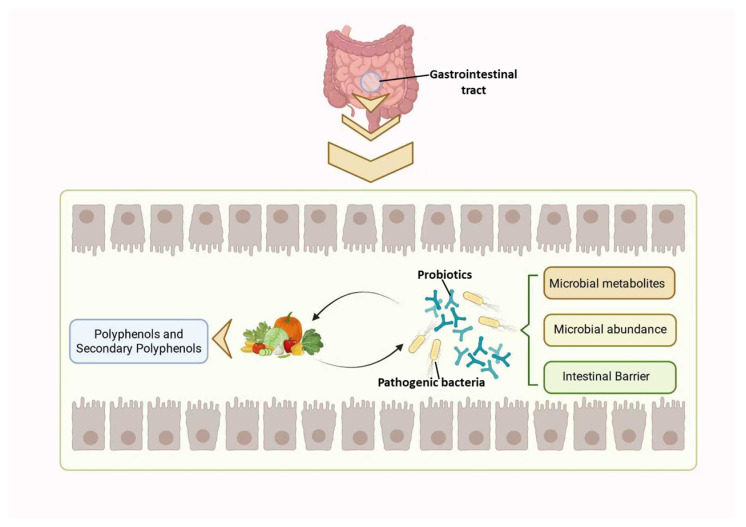
Interaction between polyphenols and gut microbiota.

## Data Availability

No new data were created or analyzed in this study. Data sharing is not applicable to this article. Data sharing is not applicable.

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
