# Peer review of "Targeting the Gut Microbiota: Mechanistic Investigation of Polyphenol Modulation of the Gut–Brain Axis in Alzheimer’s Disease"

_ijms, 2026, doi:10.3390/ijms27020604_

Round 1

Reviewer 1 Report

Comments and Suggestions for Authors

The manuscript addresses an important and rapidly developing research area—the potential of polyphenols to modulate the gut–brain axis and their possible application in the prevention or treatment of Alzheimer’s disease (AD). While the topic is highly relevant, the current version of the manuscript requires substantial revision to improve its clarity, structure, and scientific accuracy. Below I provide specific comments that, in my opinion, would significantly strengthen the work.

  1. Title–content Mismatch. The title suggests a clear emphasis on the role of polyphenols in modulating the gut–brain axis for the Prevention and Treatment of Alzheimer’s Disease. However, the manuscript does not sufficiently articulate the connection of Prevention and Treatment of AD. The proposed mechanistic links should be highlighted more explicitly throughout the text.
  2. Abstract structure. The abstract presents condensed literature data but does not specify the aims, objectives, or methodological approach of the review. These elements should be added to provide a clearer understanding of the manuscript’s scope.
  3. Structure and clarity of Section 2. Section 2 is difficult to follow. The text would benefit greatly from clearer paragraph separation, especially where different polyphenolic compounds are discussed (e.g., Lines 106, 114, 138, 173). Alternatively, the authors may structure the section into subsections, similar to what they have already done within Section 3.
  4. Abbreviations. Numerous abbreviations are used without prior definition (APP, MDA, AA, KYNA, COX, LOX, AChE, GABAA, HPA, HHDP, etc.). All abbreviations should be defined upon first use.
  5. Myricetin concentration. In line 112, it would be helpful to specify the exact concentrations of myricetin being discussed.
  6. Mechanism of quercetin action. In line 161, the statement that quercetin affects p53 to reduce ROS production is vague and sounds speculative. Please clarify the proposed mechanism.
  7. Consistency in defining polyphenols. At line 66 the authors claim that polyphenols contain at least two benzene rings, yet at line 174 they discuss ferulic acid, which contains only one benzene rings. This inconsistency requires clarification.
  8. Missing term “macrophages.” The phrase “undergo chronic M1 polarization” (line 195) lacks the word “macrophages,” which is necessary for clarity.
  9. Proportion of microbiome affected. At line 244, the authors state that specific components of the microbiota undergo changes. Please indicate the approximate percentage or proportion affected among all the microbiome.
  10. Paragraph separation around line 272. The text beginning at line 272 should be separated into a new paragraph.
  11. Mechanistic explanation of SCFA effects. Lines 276–279 contain a general statement regarding SCFAs regulating host-microbiota metabolic processes via GPCRs or histone deacetylase pathways. This requires explanation, as the current wording appears overly broad and somewhat speculative.
  12. Reorganization of sections. The manuscript structure would improve if Section 12 (“The Link Between Polyphenols and Alzheimer’s Disease”) were moved after Section 3 (“The Gut Microbiota-Gut-Brain Axis and Its Connection to Alzheimer's Disease”). Section 3 is logically organized and would better contextualize Section 2, which describes the effects of exogenous compounds in AD. This order also aligns with the structure of the Introduction.
  13. Unclear abbreviation “4AD.” The meaning of the abbreviation “4AD” in the title of Section 4 is unclear and should be explained.
  14. Commensals in SPARSE zones.
    Please clarify whether the number of commensal microorganisms in SPARSE zones changes during AD and whether polyphenols influence this phenomenon.
  15. Differential effect on enzymes. Lines 541–544 mention the influence of polyphenols on several bacterial enzymes. It would be useful to explain why similar effects are not observed on enzymes of eukaryotic gut cells.
  16. Critical evaluation of current knowledge. The Discussion section should include a critical assessment of current understanding of polyphenols and the Gut–Brain Axis in AD. It would be valuable to discuss which mechanisms are well established and which knowledge gaps are most urgent to address.
  17. Authors’ expertise. The manuscript would benefit from a brief statement clarifying the authors’ expertise in this field.
  18. Figures and schematic summaries. To improve readability and comprehension, I strongly recommend adding figures or schematic illustrations. A summary diagram showing the etiological factors in AD, the involvement of the gut–brain axis, and the points at which polyphenols exert their influence would significantly enhance the manuscript.

Overall Recommendation: Major Revision. The manuscript addresses an important topic and has the potential to contribute meaningful insights, but substantial reorganization, clarification, and expansion are required before it can be considered for publication.

Comments on the Quality of English Language

The English could be improved to more clearly express the research.

Author Response

Thank you for your valuable peer review comment. We have carefully incorporated and implemented the suggested revisions, and have prepared a detailed response outlining the modifications. Please refer to the accompanying document for specific changes.

Reviewer 2 Report

Comments and Suggestions for Authors

Overall, the paper is of good quality, but it requires several minor revisions and improvements. One notable gap is the absence of figures. I recommend including at least two original figures, strategically placed within the manuscript to enhance clarity and visual engagement, as well as one figure as a summary at the beginning of the paper.

The citations are relevant and up to date; however, some references are redundant. For example, multiple papers are cited for basic Alzheimer's disease prevalence statistics, and certain subsections are over-referenced, with more than 20 citations supporting simple background statements.

I would also like to highlight several instances of potentially misleading phrasing. For example, the statement “Polyphenols can precisely regulate gut microbiota” should be softened, as their effects are significant but not “precise”. In addition, certain mechanistic claims are presented more definitively than current evidence supports, particularly regarding bidirectional microbiota interactions and direct modulation of Alzheimer’s disease biomarkers. These statements should be qualified using more appropriate scientific language, such as: may, suggest, or emerging evidence indicates to avoid overstating the strength of available data.

To elevate the manuscript to a higher level, improvements in text organization are also recommended. Several paragraphs are extremely dense and tend to combine results from numerous studies without adequate synthesis, making it difficult for readers to identify the main conclusions. Some sections—especially the nanodelivery portion—contain repetitive structures (“X nanoparticles improved Y, reduced Aβ, suppressed inflammation…”), which affects readability and narrative flow. Changing these sections and providing clearer comparative analysis would strengthen coherence and impact.

The “Discussion” section seems more like a summary of findings than a structured analysis. It would benefit from clearer organization, with clear organization: research gaps, limitations, and future directions. I also recommend shortening overly long paragraphs to improve clarity and flow.

Comments on the Quality of English Language

There are several issues with the quality of the English language. The manuscript seems to have translation-style writing, and the language should be thoroughly reviewed, corrected, and improved. Specific issues include incorrect abbreviation definitions—for instance, the gut–brain axis is incorrectly abbreviated as BGS instead of GBA—and sentences with unclear antecedents, such as “It also effectively increases the abundance…”. Additional errors include incorrect plural forms (for example, “microbiota are” instead of “microbiota is”), missing commas, tense inconsistencies, and occasional typographical issues such as “fla-vanols.” These should all be corrected before publication.

Author Response

Thank you for your insightful peer review comments. We have meticulously integrated the recommended revisions and compiled a comprehensive response detailing the specific modifications. Please refer to the attached document for the delineation of changes.

Round 2

Reviewer 1 Report

Comments and Suggestions for Authors

The authors have fully addressed all the comments and recommendations raised during the review process and have introduced appropriate revisions to the manuscript. The changes are adequate, well justified, and significantly improve the overall quality of the paper. Therefore, I am satisfied with the revised version of the manuscript and consider it suitable for publication.

Author Response

Thank you very much for your recognition and encouragement regarding the manuscript revision. Your expert insights throughout the peer review process have provided invaluable guidance for the refinement of our paper.

We are delighted to learn that you are satisfied with the revised manuscript and consider it suitable for publication. Your thorough evaluation and constructive feedback have not only enhanced the academic rigor of the submission but also significantly contributed to our research articulation and logical argumentation. We sincerely appreciate the time and expertise you have dedicated to this review.